# Activities Carried Out during the First COVID-19 Lockdown by Italian Citizens

**DOI:** 10.3390/ijerph20053906

**Published:** 2023-02-22

**Authors:** Sara Paltrinieri, Barbara Bressi, Elisa Mazzini, Stefania Fugazzaro, Ermanno Rondini, Paolo Giorgi Rossi, Stefania Costi

**Affiliations:** 1Physical Medicine and Rehabilitation Unit, Azienda Unità Sanitaria Locale-IRCCS di Reggio Emilia, 42123 Reggio Emilia, Italy; 2Public Health Sciences PhD Program, Department of Clinical Sciences and Community Health, University of Milan, 20122 Milan, Italy; 3PhD Program in Clinical and Experimental Medicine, Department of Biomedical, Metabolic and Neural Sciences, University of Modena and Reggio Emilia, 41124 Reggio Emilia, Italy; 4Scientific Directorate Hospital Network, Azienda Unità Sanitaria Locale-IRCCS di Reggio Emilia, 42123 Reggio Emilia, Italy; 5Lega Italiana Contro i Tumori-LILT Reggio Emilia, 42123 Reggio Emilia, Italy; 6Regional Center for Education in Health Promotion-Luoghi di Prevenzione, 42122 Reggio Emilia, Italy; 7Epidemiology Unit, Azienda Unità Sanitaria Locale-IRCCS di Reggio Emilia, 42122 Reggio Emilia, Italy; 8Department of Surgery, Medicine, Dentistry and Morphological Sciences, Università di Modena e Reggio Emilia, 41124 Modena, Italy

**Keywords:** COVID-19, SARS-CoV-2, confinement, pandemic, quarantine, work, leisure activities, human activities, resilience, occupations

## Abstract

The SARS-CoV-2 pandemic has altered how citizens engage in activities. This study describes the new activities citizens engaged in during the first lockdown, factors that helped them cope with the confinement, the supports they used the most, and which supports they would have liked to receive. This cross-sectional study consists of an online survey made of 49 questions that was completed by the citizens of the province of Reggio Emilia (Italy) from 4 May until 15 June 2020. The outcomes of this study were explored by focusing on four of the survey questions. Of the 1826 citizens who responded, 84.2% had started new leisure activities. Males, participants who lived in the plain or foothills, and those who experienced nervousness engaged less in new activities, while those whose employment status changed, whose lifestyle worsened, or whose use of alcohol increased engaged in more activities. The support of family and friends, leisure activities, continuing to work, and an optimistic attitude were perceived to be of help. Grocery delivery and hotlines providing any type of information and mental health support were used frequently; a lack of health and social care services and of support in reconciling work with childcare was perceived. Findings may help institutions and policy makers to better support citizens in any future circumstances requiring prolonged confinement.

## 1. Introduction

“Occupations are the everyday activities that people do as individuals, in families and communities to occupy time and bring meaning and purpose to life” [1]. Engaging in occupations is fundamental to pursuing our individual needs; the SARS-CoV-2 pandemic, however, has forced us to change or adapt these activities.

The SARS-CoV-2 virus continues to spread worldwide, with more than half a billion cases documented up to October 2022 [2]. The most frequent symptoms of COVID-19, the disease caused by the virus, are fever, cough, shortness of breath, and fatigue [3]. Infection severity can be mild or moderate but can also require intensive acute care and hospitalization. The long-term effects of COVID-19 are being investigated, and persistent symptoms and restrictions in participation in daily life have been described [4,5,6,7].

Since the beginning of the pandemic, intermittent periods of restrictions on freedom of movement (lockdowns) have been imposed on populations living in areas where the risk of contagion is greater. These measures have been enacted to halt the spread of the virus and to limit the impact on healthcare systems. In Italy, the first lockdown started on 11 March 2020, and it was especially severe: indoor and outdoor recreational activities were suspended, and schools, universities, and businesses were closed, except those providing essential services (i.e., food, municipal services, etc.). Italian citizens were allowed to go for a walk only within a radius of 200 m from their homes [8]; most restrictions lasted until the end of May 2020. After that first lockdown, less severe restrictions were imposed in autumn-winter 2020 and spring 2021. Until 1 May 2022, adult citizens without the European Union Digital COVID Certificate, the so-called “COVID pass”, could not go to their workplace or do indoor recreational activities.

COVID-19 struck out of the blue, causing high levels of fear and disbelief everywhere. The Italian population saw its habits and lifestyle suddenly change [9,10,11], with engagement in usual occupations severely impacted. As a matter of fact, the SARS-CoV-2 virus, through the lockdown measures implemented to contain it, influenced the daily routine of individuals, leading to occupational disruption, a temporary condition due to the occurrence of significant life events, diseases, or environmental circumstances [12,13]. As was seen, the imposed confinement and the fear of contagion affected social interactions and involvement in activities of daily living [14,15]. Additionally, the rate of individuals working from home drastically increased, influencing social roles and healthy lifestyles [11,16].

The first lockdown therefore reduced the fit between the person, the environment, and the occupation (PEO), the components that, according to this model, shape the individual’s occupational performance, that is, the “dynamic experience of a person engaged in purposeful activities and tasks within an environment” [17]. From a transactional perspective, the person, the environment, and the occupation are interconnected and function through their mutual relationship, which changes over time [18]. At a given time-point in life, these components can be balanced if all the occupations of meaning are mixed and carried out within an environment in a way that is congruent with a person’s abilities to perform them [17,19]. In response to a disruptive circumstance posed by the environmental context, habitual occupations may change in the type and manner in which they are carried out in order to keep the daily routine in balance; however, these changes may not always be successful. During the first lockdown, parents carried out several occupations (i.e., childcare, remote work) by adapting their activities, time, and space as a new way of staying together at home. Nevertheless, the fit between the person, the environment, and the occupation may have been minimized, as in the case of a small home environment that does not allow the subdivision of the space and activities and, therefore, the ability to effectively perform occupations associated with their roles [20]. Thus, these imbalances expose the individual to the risk of not having his/her occupational needs satisfied, emphasizing the state of occupational disruption.

Since engagement in occupations is essential in life [21], individuals had to adapt their daily routines during lockdown, and some probably started new activities to cope with the new reality [22]. However, how individuals dealt with the situation depended on their resilience, individual attitudes, roles, activities, and, last but not least, the environment (physical, social, cultural, and political) [17].

This study explored how citizens living in an Italian province, among those provinces most affected by the first peak of the COVID-19 pandemic, dealt with lockdown by engaging in occupations.

The main aim of this study was to describe the new activities individuals started during the first lockdown. Secondary aims were to (a) describe the factors that helped individuals to overcome the challenges of the lockdown, (b) investigate which types of support provided to the community were used the most in order to positively cope with the new circumstances, and (c) investigate which kinds of support the citizens would have liked to receive.

## 2. Materials and Methods

### 2.1. Research Design and Context

This cross-sectional online survey was endorsed by the Local Health Authority (LHA) of the province of Reggio Emilia (Italy). This province is in the Emilia-Romagna region, in northern Italy, and has a population of 533,158 citizens, 66% of whom are between the ages of 18 and 70. The province of Reggio Emilia registered almost 5000 SARS-CoV-2 infections on 1 May 2020, ranking it first in the region in terms of the number of cases [23].

### 2.2. Participants

Adult citizens (aged ≥ 18) living in the province of Reggio Emilia could complete the survey without restrictions and in complete anonymity between 4 May 2020 and 15 June 2020. Participants ticked a box to declare that they have read the information relating to data processing and that they agreed to participate in the study. Informed consent was obtained from all participants involved in the study.

### 2.3. Data Collection

The survey was developed in Italian by a multidisciplinary team composed of rehabilitation professionals, physicians, and epidemiologists of the LHA of Reggio Emilia. The survey was translated into English for an international readership (Appendix A). It consisted of 49 questions organized into five sections of interest: (a) sociodemographic data (14 items); (b) work-related data (5 items); (c) digital skills and availability of devices (3 items); (d) lifestyle and health status (23 items); (e) use of local support services (3 items); (f) emotional state (1 item).

As the survey was completely anonymous, the Local Ethics Committee Area Vasta Emilia Nord deemed approval unnecessary. Nonetheless, the survey underwent an ethics review conducted by an expert in the field to ascertain its appropriateness. The expert suggested eliminating a question in which we asked whether the participant was earning an income, as employment status had already been investigated (Question 17). Furthermore, Question 22 was rephrased as, before the expert’s review, we had asked whether the participant had sufficient digital skills, implying a judgment concerning which skills were considered sufficient. The other questions were judged suitable.

Before its dissemination, the survey was piloted on a convenience sample of five adults of different ages (range 24–68) and sex (three were males). However, due to the time constraints associated with the lockdown at that time, it did not undergo formal validation. The survey was then disseminated on the websites and social media of the LHA of Reggio Emilia and by the local patient associations that joined the initiative, the major municipalities of the province, and the network of the municipal pharmacies.

The outcomes of this study were explored only by three multiple choice questions (44, 46, and 48) and one open-ended question (47). These numbers refer to the survey questions, as reported in the Appendix A.

Question 44 “What new activities you are doing now that you did not do before the COVID-19 lockdown?” investigated the primary outcome of this study. The associated answer options were: reading; resting; cooking; watching TV; using social media; dedicating time to family activities; gardening/housecleaning; volunteer work; none; other (open-ended option).

The secondary outcomes were investigated through the following questions:
-48 “What/who helped you to overcome the challenges of lockdown?” The associated answer options were: family or friends; volunteers/neighbors; healthcare professionals; the municipality or local associations; leisure activities (i.e., reading, gardening, etc.); continuing to work; optimistic attitude; physical activity, drinking and/or smoking; taking care of pets; salary; drugs; other (open-ended option).

For item 48, the kinds of help reported were categorized according to the components of the PEO model [17].
-46 “Have you used/are you using any of the support made available by the municipality, the nonprofit volunteer associations, and/or the Local Health Authority?” The associated answer options were: grocery delivery; home delivery of drugs; information hotline; mental health hotline; economic aid (i.e., food vouchers); area animal care facilities; activation of social services; other (open-ended option).

Finally, by way of open-ended question 47 “What kind of support would you have liked in order to deal with this situation but which was not available?”, we sought information about the local support services that citizens would have liked during the lockdown but which were not available.

This report was drafted according to the STROBE Checklist (Appendix A). The study was prospectively registered on CliniclTrial.gov (ID of the registered studyNCT04423978).

### 2.4. Data Analyses

Descriptive statistics were used to illustrate the characteristics of the respondents to the survey, i.e., sex, age, education level, household, presence of children, home size, area of residence (plain: northern area; center: city center and cities less than 20 km away; foothills: southern area), changes in employment, changes in lifestyle, changes in alcohol consumption, availability of internet connection and electronic devices, level of digital skills, and psychological distress.

We described the rate of participants who engaged in new activities during lockdown (no new activities, from one to two new activities, and from three to eight new activities). We then performed separate multinomial-logistic regression models, adjusted for sex, age, and education level, calculating odds ratio (ORs) and the relative 95% confidence intervals (CIs) to observe the associations between each independent variable separately and the number of new activities participants engaged in. Analyses were performed using Jamovi (version 1.6) [24].

We described the number and types of new activities engaged in, the factors (who and what) that helped the individuals, and the local support services used. Proportions were calculated on the total number of participants, and data are graphically reported.

The local support services that participants would have liked during lockdown, but that were not available, are summarized in homogeneous categories, and the proportions of participants that mentioned any category were calculated on the total number of participants who answered this question and are graphically reported.

## 3. Results

The online survey was completed by a total of 1826 participants, whose main characteristics are reported in Table 1.

Females accounted for 76.5% of the sample, 88.7% were adult or middle-aged, and 92.8% had a medium/high level of education. The majority of the participants lived with at least one other person (88.6%), in homes >100 m^2^ (48.2%), located in the city center (76.5%). Parents represented 58.4% of the sample, 35.7% of whom had children over the age of 12. More than half of the participants were in paid employment, of whom 37.5% did not change employment status during the lockdown, while 29.8% started more remote work, and 5.6% stopped working. More than 60% of participants reported good or excellent digital skills.

During lockdown, 39.3% (n. 718) engaged in few new activities (one or two), while 46.9% (n. 856) engaged in several new activities (from three to eight).

Figure 1 describes the new activities engaged in during the first lockdown. These new activities were cooking, reading, and watching TV, performed by 43.6%, 38.7%, and 35.4% of participants, respectively, followed by gardening, housecleaning, and rest.

Table 1 reports sociodemographic data, household composition, changes during lockdown, technology, and psychological distress associated with the number of new activities engaged in during lockdown. Males seemed to engage less than females in several new activities (OR 0.58, 95% CI 0.39–0.85) and participants residing in the plain or the foothills of the province seemed to engage in fewer new activities than those living in the center (plain: OR 0.60, 95% CI 0.36–0.99; foothills: OR 0.45, 95% CI 0.24–0.86, respectively).

Participants who underwent changes in work during the lockdown seemed to engage in more new activities than those whose employment condition did not change (more remote work: OR 1.89, 95% CI 1.25–2.86; suspended work: OR 3.61, 95% CI 1.38–9.49).

Similarly, participants that reported a worsening in lifestyle (OR 1.46, 95% CI 1.01–2.12) or an increase in alcohol consumption (OR 2.23, 95% CI 1.18–4.19) seemed to engage in more new activities than those not reporting any changes in lifestyle. Finally, nervousness seemed to be the only symptom of psychological distress that was associated with less engagement in several new activities (OR 0.66, 95% CI 0.44–0.99).

Figure 2 describes the factors that helped individuals to cope with the challenges of lockdown, all of which were encompassed by the three components of the PEO model, namely person, environment, and occupation. However, environment was the most widely represented component of the model, as 59% of participants reported that family/friends and the institutional environment (1.7% municipality/local associations, and 1.6% healthcare professionals) helped them to cope. Regarding occupations, leisure activities and continuing to work were helpful for 55.4% and 46.9% of the participants, respectively, followed by salary (40.1%), physical activity (32.7%), taking care of pets (19.75%), and volunteer work (2.8%). Finally, regarding the person component, which includes the self-concept of optimistic attitude, this was reported as helpful by 39.6% of participants. Additional personal factors that were deemed helpful were smoking and/or alcohol drinking (4.1%) and drugs (2.2%).

Figure 3 describes the local support services used by 476 participants. Grocery delivery and hotlines providing information and mental health support were the most used, as reported by 18.9% and 6.6% of participants, respectively.

Finally, 176 participants (9.6% of the whole cohort) would have used some other support services if they had been available in their community: 20.5% said they would have like support in reconciling working remotely and having children at home, while 15.3% said they would have liked more health and social care support (Figure 4). Healthcare support referred both to those individuals with COVID-19 (quarantine, swabs, distancing rules, etc.) and to those who needed ongoing care for diseases other than COVID-19, such as emergency medical services, access to the general practitioner, and access to specialized diagnostic or therapeutic services (i.e., computed axial tomography, rehabilitation, asthma center). Additionally, several participants stated the need to maintain access to disability day care centers and more support to manage frail family members (i.e., elderly, disabled, with chronic diseases) at home. Moreover, the need for support in managing insomnia, quitting smoking, and relationships with household members or other unspecified psychological issues was also expressed.

Further, 14.1% of participants stated they would have liked a grocery delivery service, and 10.7% applied for economic aid. Few individuals stated they would have liked clear but less stressful information (7%), relaxing programs on television (1%), a list of the businesses open in the community (0.5%), and donation collection points.

## 4. Discussion

Data from this survey suggest that citizens of the province of Reggio Emilia may have engaged in new activities during the first lockdown. However, participants who took part in this survey cannot be considered representative of the population of the province of Reggio Emilia, which represents a restricted geographical area in Italy. Compared to demographics of this province, in our sample females (76.5% versus 50.8%), individuals of working age (88.7% versus 61%), and individuals living with at least one person (88.6% versus 64%) were over-represented [25]. In line with our results, the over-representation of females, of individuals of working age, and of those who live with at least one person also occurred in other surveys disseminated during the first lockdown in the city of Reggio Emilia, [26], in Denmark [27], and in Canada or the United States [28].

Although the lack of representativeness of the sample is a major limitation, data from this survey suggest that a wide range of new activities were undertaken by Italian citizens during the first lockdown, with cooking, reading, and watching TV being the most practiced. The new leisure activities engaged in were mostly quiet, as home isolation and the impossibility of leaving the house except for short distances limited the types of new activities that could be engaged in. Similarly, watching TV was the most prevalent activity among German citizens during the lockdown [29], and in Belgium more than 97% of respondents continued their indoor activities, while outdoor activities (i.e., sport) were performed indoors or were suspended [30].

In our sample, males seemed to have engaged in fewer new activities than did females, who reorganized their daily routine to incorporate them. This could be explained by looking at previously published data on this same cohort, which show that men were less affected by suspension of work activities (only 14% of males versus 86%of females); most of the male participants were able to continue working remotely or at their usual place of employment. Thus, it is likely that males had less time to engage in a wide range of new activities.

This aspect may have caused high levels of stress and burnout [31]. Congruently, involvement in meaningful activities in addition to employment appeared to be beneficial to the mental health of citizens, as reported by Cruyt (2021). Immediately after the announcement of the first lockdown, the Italian authorities paid considerable attention to the psychological distress that could result from the confinement, informing the population on how to maintain positive lifestyles through mass media and a psychological support hotline [32]. This attention to lifestyles was likely necessary, as in stressful situations people increase their use of alcohol, nicotine, and drugs, as also demonstrated by our data. To be highlighted, nicotine, alcohol, and drugs can increase the risk of contracting the virus due to the activation of ACE2 receptors and the weakening of the body’s barrier functions [33,34,35].

Additionally, citizens who did not live in city centers seemed to have performed fewer new activities within the home environment; it is possible that they carried out more outdoor activities due to the presence of more green or wooded areas in their immediate neighborhood.

Participants who changed their habitual activities of daily living, such as physical activity and employment, engaged in many leisure activities to readjust and to achieve a balance in their new daily routine. Even if we do not know whether this readjustment was beneficial to participants in the present study, filling time over lockdown by learning new activities was perceived positively by English citizens who started or increased working remotely during the pandemic [36].

Our survey data show that family and friends were the main source of support to preserve psychological health, as other studies have also shown [37]. Despite physical distancing, relationships and interactions were fundamental to coping with difficulties and mood disturbances [38]. For those family members not living under the same roof, relationships were maintained through social media and electronic devices. These virtual relationships, however, require devices and a stable internet connection, which 95% and nearly 90% of participants, respectively, had. In addition, 65% of participants stated they had good or excellent digital skills, with only a small proportion of participants (5.6%) stating they would have liked more support related to virtual connections.

Besides the crucial support of family and friends, aided by virtual communication, the two other pillars that sustained individuals during lockdown were leisure activities and being able to continue to work. In addition to providing income, employment represents the basis for a structured daily routine, which facilitates the management of all meaningful occupations and improves life satisfaction [39,40]. The lockdown led to important changes in employment, namely a reduction in work tasks and work schedule, remote work, and, in certain conditions, the loss of employment [11]. The latter scenario can be detrimental, as it leads to financial hardship and emotional distress. In fact, as stated by Escudero-Castillo et al., Spanish citizens who lost their job seemed to have the worst psychological consequences as compared to those who continued to work during the lockdown [41]. Moreover, those who worked from home perceived less well-being than those who continued to work at their workplace [42]. Congruently, 20.5% of participants in our survey reported the lack of support in reconciling remote work and parenting at home. During the lockdown, the roles of worker and parent, which usually occupy distinct times and places, overlapped due to the closure of schools and to the increase in remote work. The home and the immediate neighborhood were the only areas where occupations could take place, thus creating possible conflicts among household members, reducing productivity [42], and limiting readjustment to the new daily routine through the engagement in new, meaningful activities [43]. In the Netherlands, hours spent in paid work decreased significantly during lockdown, and families with young children seemed to have spent more time in preparing meals, childcare, and leisure activities (i.e., relaxing) [15]. Although Dutch males seemed to participate more in the household activities, household management may have fallen to the female sex, who are more frequently responsible for childcare and housework [15,43].

Additionally, having an optimistic attitude was a strong basis for dealing with the confinement, as stated by 41% of the sample investigated. This personal characteristic seemed to protect against anxiety and depression, as also confirmed by other research [44].

With regard to support services, grocery delivery was used by 18.9% of participants.

Unlike other realities, where this service was used by the most fragile or infected individuals, as reported in German and America studies [45,46], the sub-group of participants to this survey who used grocery delivery were primarily adult or middle-aged individuals (90.5%), living with at least one person (93.6%), with no chronic diseases (82%). Thus, fragility may not explain such behaviors, while feelings of psychological distress probably do, as more than a half of the users of support services perceived uncertainty at the time of the survey.

Nevertheless, it should be noted that about three-quarters of the respondents to our survey did not use this service, perhaps because going to the supermarket was one of the few exceptions allowed to confinement and was thus a way to get some fresh air. Furthermore, even though receiving information through hotlines was appreciated, citizens would have preferred to receive clearer information on the pandemic. This perception was similar to that of Swiss citizens, who claimed they had difficulty assessing the reliability of information relating to COVID-19 during the first lockdown [47].

Finally, our participants perceived a lack of support from health and social care services in addressing issues related to COVID-19 or other diseases. A reduction in the use of health services was reported worldwide [48]. Both in Europe and in the United States, patients with chronic conditions reported a lack of consultations with their physicians [49], while a decrease in accessing the emergency department was documented in Italy [50]. Additionally, the Spanish population’s satisfaction with its healthcare system decreased between the first and second waves of pandemic [51]. Overall, healthcare systems worldwide had to reorganize their provision of services to face the pandemic; the citizens’ opinions were likely strongly influenced by this reorganization.

### Strengths and Limitations

One of the strengths of the study is that, to our knowledge, this is the first time that the PEO model described by Law (1996) has been applied to investigate how people’s occupations may have been altered due to the restrictions imposed by the lockdown. Nonetheless, these results can also be useful for learning about how restrictions on the freedom of movement might affect the occupations of individuals in the community or behavioral patterns that could arise in the future in case of the need to limit social contact, even in the absence of formal constraints.

As stated before, participants who took part in this survey cannot be considered representative of the population of the province of Reggio Emilia. Even though the online survey was open to all adult individuals living in the province, the institutional channels used to disseminate the survey led us to intercept a sample consisting predominantly of adult or middle-aged female individuals with a medium or high level of education and a high level of digital skills. The high percentage of females reflects the risk of a self-reporting bias, which may have influenced the results. As a matter of fact, it is possible that in this province females are more likely to be engaged in occupations that concern home productivity (i.e., cooking, housecleaning), including childcare, as also reported in the results of Yerkes et al. (2021). Furthermore, citizens with poor digital skills or without devices/IT connection were underrepresented. This could be due to the intrinsic characteristics of the survey, which could only be filled out online. We can hypothesize that these individuals without appropriate technology may have had more difficulties and were therefore more vulnerable from societal, financial, and occupational perspectives. As a consequence, we do not know how these more vulnerable citizens reorganized themselves during the lockdown and what support they would have liked the most to deal with the situation.

Moreover, for feasibility reasons, most of the questions posed were multiple choice; although an open option was always available, the list of answers was broad but not unlimited. Thus, it is possible that the proposed list of choices limited the description of other relevant occupations engaged in by participants during lockdown. Concerning question 44, we provided a list of new activities that participants could have engaged in at home, as a way to react to the restrictions on their freedom of movement. However, some choices represent activities that an individual usually carries out as part of a daily routine (i.e., cooking, housecleaning), regardless of lockdown. Therefore, it is possible that some participants interpreted the question considering the new activities as occupations they performed more frequently or about which they had learned something new, for example, a new recipe.

In addition, we underline that our survey did not investigate the social relationships between cohabitants and with neighbors. This would have provided insights into the facilitating or hindering dynamics of daily living during confinement.

Furthermore, considering that environmental factors play a central role in determining how individuals engage in occupations, another limitation of this study is due to the poor generalizability of our results in countries where the provision of services has been reorganized differently in terms of typology and accessibility.

Finally, we emphasize that this survey did not employ validated tools, an aspect that should be considered when interpreting our results.

## 5. Conclusions

The first lockdown quickly and dramatically disrupted the daily life of people all over the world. Thus, changes in meaningful occupations occurred during the lockdown to meet individuals’ needs and their adjustment to a new daily routine. We identified three main pillars that citizens used to achieve this: environment, occupations, and personal factors, namely the support of family and friends and community resources, leisure activities, being able to continue to work, and a self-optimistic attitude. These results might help institutions and policy makers to support citizens in any future circumstance requiring prolonged confinement.

From a public health perspective, more useful information and advice that encourage people to stay involved in their meaningful occupations could be disseminated. Moreover, the proposed PEO model may guide the development of tools to investigate whether citizens are appropriately addressing the three main pillars in any future moment of great difficulty. Furthermore, healthcare professionals and hotlines that provide individuals with specific support could investigate whether there are any difficulties performing occupations. In this case, specific support or services might be activated.

Finally, it is possible that there were still imbalances between environment, occupations, and personal factors in the most vulnerable individuals, for example, those who had lost their employment. For this reason, future research should investigate the aftermath of the pandemic, especially for vulnerable individuals who may not have balanced their occupations during lockdown and thus not adapted to the new circumstances.

## Figures and Tables

**Figure 1 ijerph-20-03906-f001:**
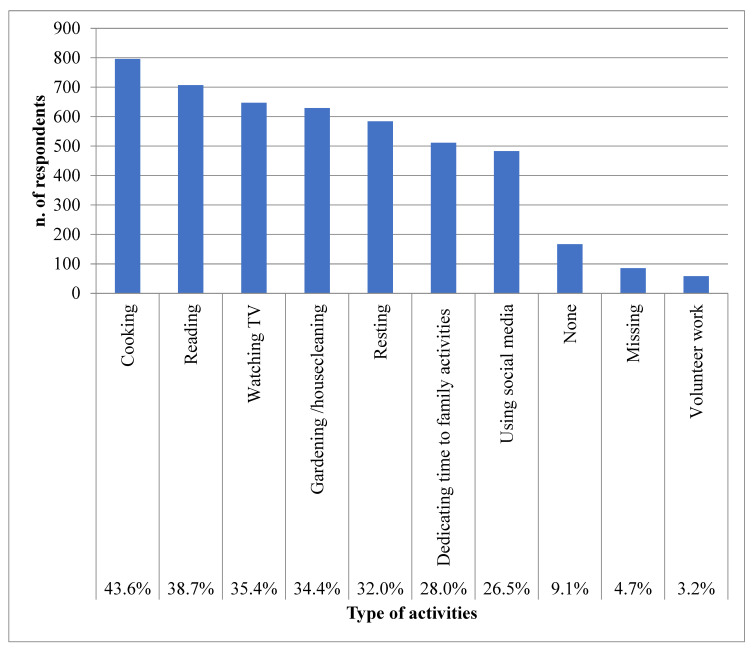
Percentages of participants who carried out new activities, described also as types, during the first lockdown.

**Figure 2 ijerph-20-03906-f002:**
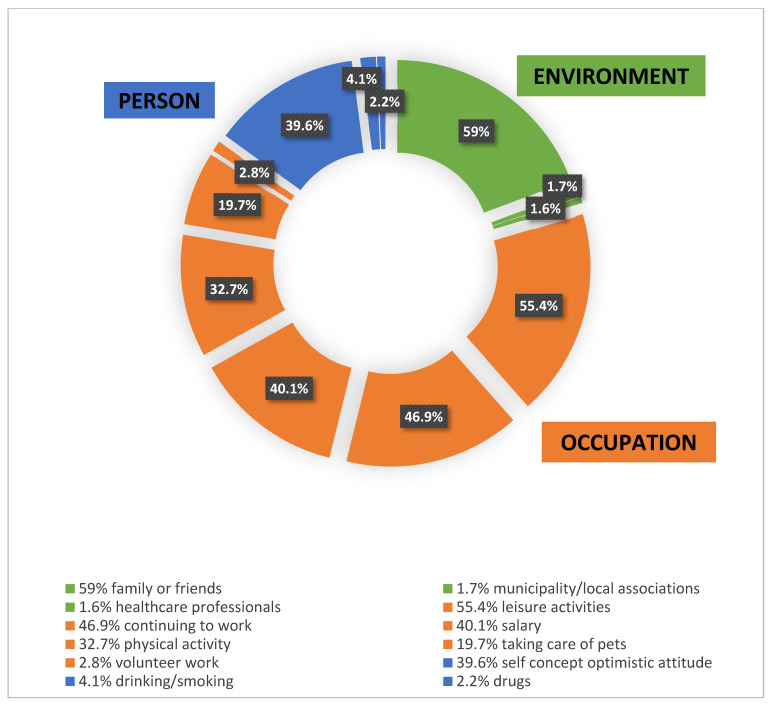
Percentages of participants who reported the kinds of factors (according to the Person, Environment, Occupation model) that helped them to deal with the first lockdown.

**Figure 3 ijerph-20-03906-f003:**
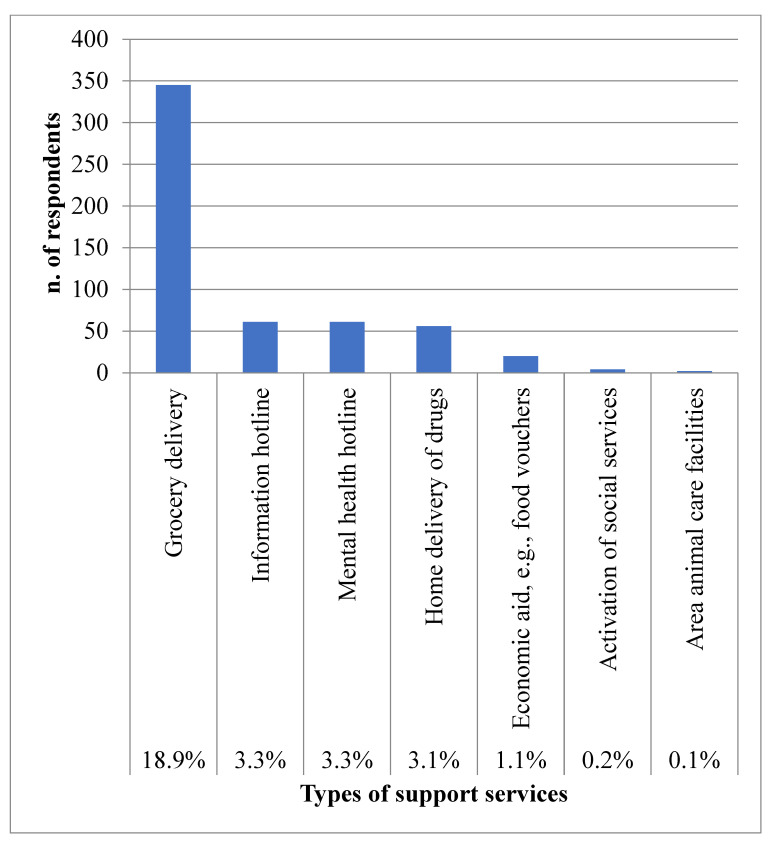
Percentages of participants who used support services, described also as types, during the first lockdown.

**Figure 4 ijerph-20-03906-f004:**
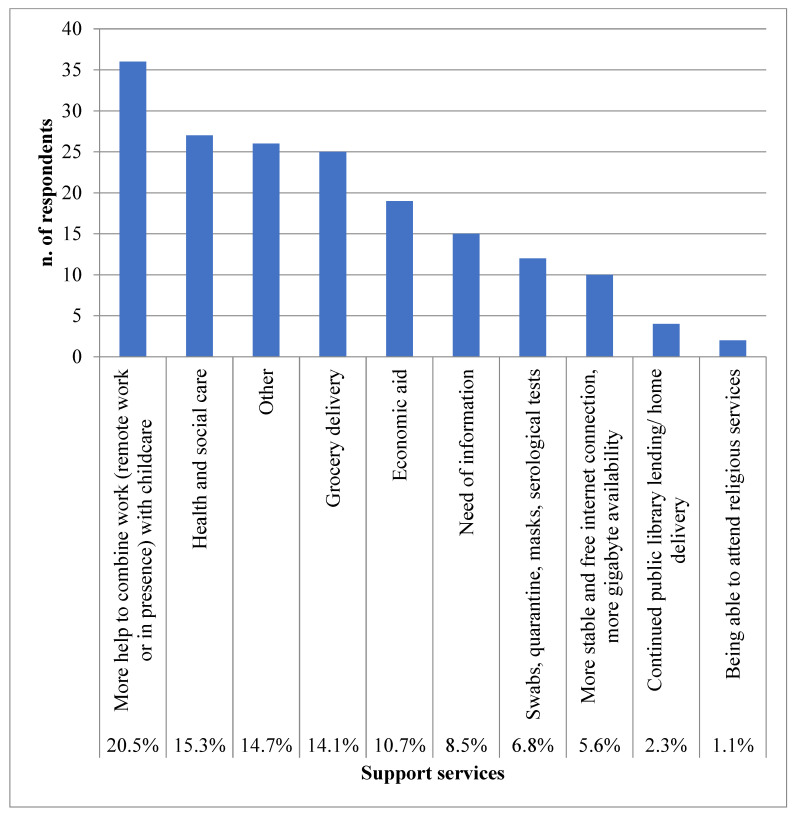
Percentages of participants who would have liked to use support services, described also as types, which were not available during the first lockdown.

**Table 1 ijerph-20-03906-t001:** Odds Ratios For Number Of New Leisure Activities (None, From One To Two, Or From Three To Eight) During COVID-19 Lockdown By Sociodemographic Data, Household, Changes During Lockdown, Technology, And Psychological Distress.

			New Leisure Activities												
			Total	None	One or Two	From Three to Eight	Missing	One or Two		95% Confidence Interval		From Three to Eight		95% Confidence Interval	
			N (%)					P	OR	Lower	Upper	P	OR	Lower	Upper
			1826 (100)	167 (9.1)	718 (39.3)	856 (46.9)	85								
Sociodemographic data	Sex	female	1397 (76.5)	120 (8.6)	524 (37.5)	697 (49.9)	56		1.00	-	-		1.00	-	-
		male	423 (23.2)	46 (10.9)	193 (45.6)	159 (37.6)	25	0.79	0.95	0.65	1.39	0.006	0.58	0.39	0.85
		missing	6 (0.3)	1 (16.7)	1 (16.7)	/	4	-	-	-	-	-	-	-	-
	Age ^†^	adults	818 (44.8)	68 (8.3)	329 (40.2)	390 (47.7)	31	-	1.00	-	-	-	1.00	-	-
		middle-aged	802 (43.9)	82 (10.2)	308 (38.4)	374 (46.6)	38	0.10	0.73	0.51	1.06	0.06	0.71	0.50	1.02
		elderly	194 (10.6)	16 (8.2)	76 (39.2)	90 (46.4)	12	0.83	0.93	0.51	1.73	0.82	0.93	0.51	1.71
		missing	12 (0.7)	1 (8.3)	5 (41.7)	2 (16.7)	4	-	-	-	-	-	-	-	-
	Education Level	low	94 (5.1)	9 (9.6)	41 (43.6)	42 (44.7)	2	-	1.00	-	-	-	1.00	-	-
		medium	805 (44.1)	59 (7.3)	301 (37.4)	406 (50.4)	39	0.79	1.11	0.51	2.43	0.41	1.39	0.64	3.03
		high	889 (48.7)	94 (10.6)	367 (41.3)	389 (43.8)	39	0.58	0.80	0.37	1.74	0.51	0.77	0.36	1.67
		missing	38 (2.1)	5 (13.2)	9 (9)	19 (50)	5	-	-	-	-	-	-	-	-
Household	Household Composition	alone	208 (11.4)	23 (11.1)	87 (41.8)	89 (42.8)	9	-	1.00	-	-	-	1.00	-	-
		at least one person	1618 (88.6)	144 (8.9)	631 (39)	767 (47.4)	76	0.66	1.12	0.68	1.84	0.29	1.31	0.80	2.16
	Children	no	700 (38.3)	65 (9.3)	289 (41.3)	318 (45.4)	28	-	1.00	-	-	-	1.00	-	-
		yes, at least one under the age of 12	415 (22.7)	38 (9.2)	158 (38.1)	206 (49.6)	13	0.80	0.95	0.60	1.48	0.73	1.08	0.69	1.68
		yes, all over age 12	652 (35.7)	62 (9.5)	255 (39.1)	300 (46)	35	0.97	1.01	0.62	1.65	0.84	0.95	0.59	1.54
		missing	59 (3.2)	2 (3.4)	16 (27.1)	32 (54.2)	9	-	-	-	-	-	-	-	-
	Home Size	<50 m^2^	57 (3.1)	6 (10.5)	23 (40.4)	26 (45.6)	2	-	1.00	-	-	-	1.00	-	-
		from 50 m^2^ to 100 m^2^	830 (45.5)	67 (8.1)	334 (40.2)	386 (46.5)	43	0.81	1.13	0.41	3.11	0.90	1.07	0.39	2.91
		>100 m^2^	880 (48.2)	89 (10.1)	336 (38.2)	420 (47.7)	35	0.80	0.88	0.32	2.40	0.88	0.92	0.34	2.51
		missing	59 (3.2)	5 (8.5)	25 (42.4)	24 (40.7)	5	-	-	-	-	-	-	-	-
	Area of Residence	center	1396 (76.5)	116 (8.3)	569 (40.8)	653 (46.8)	58	-	1.00	-	-	-	1.00	-	-
		plain	211 (11.6)	25 (11.8)	76 (36)	99 (46.9)	11	0.04	0.60	0.36	0.99	0.07	0.64	0.39	1.03
		foothills	127 (7)	15 (11.8)	35 (27.6)	71 (55.9)	6	0.01	0.45	0.24	0.86	0.34	0.75	0.41	1.37
		na	71 (3.9)	8 (11.3)	31 (43.7)	26 (36.6)	6	0.72	0.86	0.38	1.93	0.31	0.65	0.29	1.49
		missing	21 (1.2)	3 (14.3)	7 (33.3)	7 (33.3)	4	-	-	-	-	-	-	-	-
Changes during lockdown	Changes in Employment	unchanged	685 (37.5)	87 (12.7)	272 (39.7)	298 (43.5)	28	-	1.00	-	-	-	1.00	-	-
		more remote work	544 (29.8)	41 (7.5)	217 (39.9)	262 (48.2)	24	0.02	1.66	1.09	2.51	0.002	1.89	1.25	2.86
		suspended	103 (5.6)	5 (4.9)	30 (29.1)	67 (65)	1	0.22	1.87	0.69	5.09	0.009	3.61	1.38	9.49
		na ^‡^	313 (17.1)	25 (8)	135 (43.1)	130 (41.5)	23	0.06	1.74	0.98	3.09	0.35	1.32	0.74	2.35
		missing	181 (9.9)	9 (5)	64 (35.4)	99 (54.7)	9	-	-	-	-	-	-	-	-
	Changes In Lifestyle	unchanged	972 (53.2)	97 (10)	404 (41.6)	418 (43)	53	-	1.00	-	-	-	1.00	-	-
		improved	97 (5.3)	9 (9.3)	26 (26.8)	54 (55.7)	8	0.29	0.65	0.29	1.44	0.53	1.27	0.60	2.69
		worse	641 (35.1)	51 (8)	247 (38.5)	328 (51.2)	15	0.47	1.15	0.79	1.67	0.05	1.46	1.01	2.12
		missing	116 (6.4)	10 (8.6)	41 (35.3)	56 (48.3)	9	-	-	-	-	-	-	-	-
	Changes in Alcohol Consumption	unchanged	1275 (69.8)	131 (10.3)	493 (38.7)	597 (46.8)	54	-	1.00	-	-	-	1.00	-	-
		increased	229 (12.5)	12 (5.2)	94 (41)	117 (51.1)	6	0.02	2.10	1.11	3.96	0.01	2.23	1.18	4.19
		decreased	231 (12.7)	19 (8.2)	91 (39.4)	105 (45.5)	16	0.49	1.21	0.70	2.09	0.43	1.24	0.72	2.13
		missing	91 (5)	5 (5.5)	40 (44)	37 (40.7)	9	-	-	-	-	-	-	-	-
Technology	Stable Internet Connection	no	130 (7.1)	11 (8.5)	61 (46.9)	51 (39.2)	7	-	1.00	-	-	-	1.00	-	-
		don’t know	18 (1)	1 (5.6)	9 (50)	8 (44.4)	0	0.68	1.58	0.18	13.80	0.66	1.63	0.18	14.52
		yes	1634 (89.5)	154 (9.4)	628 (38.4)	784 (48)	68	0.40	0.75	0.39	1.47	0.78	1.10	0.56	2.17
		missing	44 (2.4)	1 (2.3)	20 (45.5)	13 (29.5)	10	-	-	-	-	-	-	-	-
	Electronic Devices	no	30 (1.6)	1 (3.3)	11 (36.7)	15 (50)	3	-	1.00	-	-	-	1.00	-	-
		yes	1737 (95.1)	164 (9.4)	688 (39.6)	813 (46.8)	72	0.38	0.40	0.05	3.14	0.30	0.34	0.04	2.59
		missing	59 (3.2)	2 (3.4)	19 (32.2)	28 (47.5)	10	-	-	-	-	-	-	-	-
	Digital Skills	bad	51 (2.8)	5 (9.8)	25 (49)	19 (37.3)	2	-	1.00	-	-	-	1.00	-	-
		sufficient	540 (29.6)	52 (9.6)	218 (40.4)	244 (45.2)	26	0.79	0.87	0.31	2.45	0.62	1.31	0.45	3.77
		good/excellent	1181 (64.7)	106 (8)	463 (39.2)	563 (47.7)	49	0.84	0.90	0.32	2.54	0.40	1.57	0.55	4.53
		missing	54 (3.0)	4 (7.4)	12 (22.2)	30 (55.6)	8	-	-	-	-	-	-	-	-
Psychological distress	Nervousness	no	1337 (73.2)	118 (8.8)	514 (38.4)	641 (47.9)	64	-	1.00	-	-	-	1.00	-	-
		yes	359 (19.7)	39 (10.9)	152 (42.3)	157 (43.7)	11	0.47	0.86	0.57	1.30	0.05	0.66	0.44	0.99
		missing	130 (7.1)	10 (7.7)	52 (40)	58 (44.6)	10	-	-	-	-	-	-	-	-
	Upset	no	1284 (70.3)	122 (9.5)	496 (38.6)	605 (47.1)	61	-	1.00	-	-	-	1.00	-	-
		yes	372 (20.4)	36 (9.7)	160 (43)	163 (43.8)	13	0.69	1.09	0.72	1.65	0.42	0.84	0.56	1.28
		missing	170 (9.3)	9 (5.3)	62 (36.5)	88 (51.8)	11	-	-	-	-	-	-	-	
	Worry	no	914 (50.1)	85 (9.3)	353 (38.6)	429 (46.9)	47	-	1.00	-	-	-	1.00	-	-
		yes	810 (44.4)	74 (9.1)	337 (41.6)	369 (45.6)	30	0.63	1.09	0.77	1.55	0.59	0.91	0.64	1.29
		missing	102 (5.6)	8 (7.8)	28 (27.5)	58 (56.9)	8	-	-	-	-	-	-	-	-
	Fear	no	1404 (76.9)	134 (9.5)	550 (39.2)	662 (47.2)	58	-	1.00	-	-	-	1.00	-	-
		yes	303 (16.6)	24 (7.9)	121 (39.9)	141 (46.5)	17	0.44	1.21	0.75	1.97	0.84	1.05	0.65	1.70
		missing	119 (6.5)	9 (7.6)	47 (39.5)	53 (44.5)	10	-	-	-	-	-	-	-	-
	Loneliness	no	1439 (78.8)	135 (9.4)	569 (39.5)	670 (46.6)	65	-	1.00	-	-	-	1.00	-	-
		yes	249 (13.6)	20 (8)	103 (41.4)	119 (47.8)	7	0.55	1.17	0.70	1.97	0.73	1.10	0.66	1.83
		missing	138 (7.6)	12 (8.7)	46 (33.3)	67 (48.6)	13	-	-	-	-	-	-	-	-
	Uncertainty	no	778 (42.6)	80 (10.3)	308 (39.6)	356 (45.8)	34	-	1.00	-	-	-	1.00	-	-
		yes	996 (54.5)	84 (8.4)	390 (39.2)	481 (48.3)	41	0.35	1.18	0.83	1.67	0.31	1.19	0.85	1.68
		missing	52 (2.8)	3 (5.8)	20 (38.5)	19 (36.5)	10	-	-	-	-	-	-	-	-

ORs were adjusted for age, sex, and education level. age is adjusted for sex and education level; sex is adjusted for age and education level; education level is adjusted for age and sex. ^†^ adults (18–44); middle-aged (45–64); elderly (≥65). ^‡^ na = participants who were retired, students, or housewives before COVID-19 lockdown.

## Data Availability

The dataset is stored by the Azienda Unità Sanitaria Locale–IRCCS di Reggio Emilia and is available upon request from the corresponding author.

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
