# Peer review of "Activities Carried Out during the First COVID-19 Lockdown by Italian Citizens"

_ijerph, 2023, doi:10.3390/ijerph20053906_

Round 1
Reviewer 1 Report
The study which is summarized in this article is a very interesting initiative, though I am not sure about its practical implications.
My comments are the following:
1. The title of the article looks very promising, but I have doubt whether it reflects the content of this article. As the authors stated, "The main aim of this study was to describe the new activities individuals started during the first lockdown". For this dependent variable they present in table 1 a whole list of adjusted associations with several independent variable. The title of the article refers to their secondary aim "to describe the factors that helped individuals to overcome the challenges of the lockdown". For that they had one question in their questionnaire (q. 48) for which the authors present a list of factors categorized by the PEO model, with very general information on the total frequencies of the answers. They could, at least, analyze it by age and sex. As for fig. 2 – it does not add any further information above what is in the text. It could have had an added value if the overlapping of the circles would be elaborated and explained.
2. As for the non-representativeness of their sample, the authors mentioned it only in the limitations of the study. As this is the major limitation, I would expect to see it as the first item in the discussion. I might assume that the province has some sociodemographic characteristics, to which the sample composition could have been compared, and discuss the possible existing situation in the province.
3. The discussion starts with: "Data from this survey suggest that most citizens of the province of Reggio Emilia engaged in new activities during the first lockdown…". In relation to the previous comment, I would suggest being much less definite with such a sentence.
4. In the paragraph which starts in row 285, in the discussion, the authors try to explain why men were engaged in less new activities than women. It is not clear why they had to guess while they have all the needed data to analyze this question and get the answer.
5. A similar comment is in relation to what is written in rows 343-4: "With regard to support services, grocery delivery was used by 18.9% of participants. This service was probably used by the most fragile…". They could analyze this question at least by age, sex, household composition, and have a deeper discussion.
Reviewer 2 Report
The manuscript of Paltrinieri and co-authors is well-written. The conclusions made in the article meet the goals set. There are several comments that should be taken into account before the manuscript is published.
1. The figure's legends should be more informative.
2. Figure 2 is too massive and not informative enough. Authors should remake the figure as a circle diagram to make it more useful for readers.
3, It should be added to the discussion that nicotine, alcohol, and drugs increase the risk of contracting the virus not only as a social but also as a physiological factor, due to the activation of ACE2 receptors and the weakening of the body's barrier functions.
Round 2
Reviewer 1 Report
The authors answered most of my comments. I have no further comments.